# A shunt pathway limits the *CaaX* processing of Hsp40 Ydj1p and regulates Ydj1p-dependent phenotypes

**Emily R Hildebrandt, Michael Cheng, Peng Zhao, June H Kim, Lance Wells, Walter K Schmidt\***

Department of Biochemistry and Molecular Biology, University of Georgia, Athens, United States

**Abstract** The modifications occurring to *CaaX* proteins have largely been established using few reporter molecules (e.g. Ras, yeast **a**-factor mating pheromone). These proteins undergo three coordinated COOH-terminal events: isoprenylation of the cysteine, proteolytic removal of *aaX*, and COOH-terminal methylation. Here, we investigated the coupling of these modifications in the context of the yeast Ydj1p chaperone. We provide genetic, biochemical, and biophysical evidence that the Ydj1p *CaaX* motif is isoprenylated but not cleaved and carboxylmethylated. Moreover, we demonstrate that Ydj1p-dependent thermotolerance and Ydj1p localization are perturbed when alternative *CaaX* motifs are transplanted onto Ydj1p. The abnormal phenotypes revert to normal when post-isoprenylation events are genetically interrupted. Our findings indicate that proper Ydj1p function requires an isoprenylatable *CaaX* motif that is resistant to post-isoprenylation events. These results expand on the complexity of protein isoprenylation and highlight the impact of post-isoprenylation events in regulating the function of Ydj1p and perhaps other *CaaX* proteins.

**\*For correspondence:** wschmidt@bmb.uga.edu

**Competing interests:** The authors declare that no competing interests exist.

## Introduction

*CaaX* proteins have roles in many biological processes, including cancer, development, aging, and parasitic growth, among others. They are defined by a COOH-terminal *CaaX* motif where *C* is cysteine, *a* is typically an aliphatic amino acid, and *X* can be one of several residues. In some publications, the *CaaX* motif is referred to as a *CXXX* motif due to its highly degenerate nature (*Kinsella et al., 1991*; *Fu and Casey, 1999*; *Young et al., 2001*; *Wright et al., 2006*).

The *CaaX* motif is commonly cited as undergoing an ordered series of post-translational modifications. The first modification is C15 (farnesyl) or C20 (geranylgeranyl) isoprenylation of the cysteine (*C*) by cytosolic isoprenyltransferases. This is followed by endoproteolysis to remove the COOH-terminal tripeptide (*aaX*). Rce1p and Ste24p are proteases that can independently mediate this activity, but the promiscuous nature of Ste24p suggests that *CaaX* proteolysis is not likely its primary duty (*Boyartchuk et al., 1997*; *Tam et al., 1998*; *Ast et al., 2016*; *Hildebrandt et al., 2016*). The isoprenylated cysteine revealed by proteolysis is carboxylmethylated by an isoprenylcysteine carboxylmethyltransferase (ICMT/Ste14p). The *CaaX* proteases and ICMT are all integral, multi-pass membrane proteins associated with the endoplasmic reticulum (ER) (*Romano et al., 1998*; *Schmidt et al., 1998*). In some instances, additional modifications occur to *CaaX* proteins. Common examples are palmitoylation of mammalian H-Ras and N-Ras, and additional proteolysis of precursors to yeast **a**-factor and lamin A. The modifications occurring to *CaaX* proteins often impact their function and/or localization, and intense interest is focused on developing therapeutic inhibitors against all enzymatic steps associated with *CaaX* protein modification (*e.g.* farnesyltransferase inhibitors; FTIs) (*Silvius, 2002*; *Winter-Vann and Casey, 2005*; *Berndt et al., 2011*).

The modifications occurring to *CaaX* proteins have largely been interrogated using relatively few *CaaX* proteins, with Ras and Ras-related GTPases, the yeast **a**-factor mating pheromone, and lamin A being the most common reporters (*Boyartchuk et al., 1997*; *Tam et al., 1998*; *Boyartchuk and Rine, 1998*; *Kim et al., 1999*; *Roberts et al., 2008*). Whereas these proteins certainly undergo the ordered set of post-translational modifications described above, several lines of evidence support the existence of a shunt pathway that yields partially modified products. Mammalian Rab38 (CAKS *CaaX* motif) is farnesylated, but not carboxylmethylated, implying that it is not cleaved (*Leung et al., 2007*). The α (CAMQ) and β (CVLS) subunits of phosphorylase b kinase (Phk) are farnesylated yet retain their *CaaX* motifs (*Heilmeyer et al., 1992*). The major form of Gγ5 (CSFL) is geranylgeranylated and also retains its *CaaX* motif (*Kilpatrick and Hildebrandt, 2007*). Additionally, mammalian annexin A2 (CGGDD) was identified in a screen for isoprenylated proteins. Its non-canonical length motif suggests that it may not be cleaved (*Kho et al., 2004*). Detailed follow-up studies of these observations have not been reported, and the relevance of interrupted processing to protein function has not been investigated.

In this study, we developed the Ydj1p *CaaX* protein as a new reporter for assessing the role of *CaaX* proteolysis and carboxylmethylation in regulating *CaaX* protein function. Ydj1p is a yeast homolog of DnaJ, an *E. coli* heat shock protein (*Caplan and Douglas, 1991*). The homology between Ydj1p and DnaJ lies predominantly in the $NH_2$ terminus (*Caplan and Douglas, 1991*); DnaJ does not possess a *CaaX* motif. Heat shock proteins generally function in protein folding, transport, and assembly (*Qiu et al., 2006*). Among its many reported roles in yeast, Ydj1p functions with Hsp70s of the Ssa class to mediate the transfer of preproteins across mitochondrial and ER membranes (*Caplan and Douglas, 1991*; *Atencio and Yaffe, 1992*; *Becker et al., 1996*). Ydj1p is also required for growth at elevated temperature (*Caplan et al., 1992*). Farnesylation of Ydj1p is specifically required for growth at elevated temperatures, the suppression of certain prions, and maturation of Hsp90 client proteins (*Caplan et al., 1992*; *Summers et al., 2009*; *Flom et al., 2008*; *Sharma et al., 2009*). The roles of proteolysis and carboxylmethylation in regulating these activities have not previously been investigated. We report here that the yeast Hsp40 chaperone Ydj1p is a *CaaX* protein that avoids post-isoprenylation proteolysis and carboxylmethylation. We also demonstrate that there are negative consequences to Ydj1p function and localization if it is forced to undergo post-isoprenylation modification. These observations expand on the complexity of protein isoprenylation and are expected to provide a better understanding of the roles of the *CaaX* proteases and ICMT in regulating the function of *CaaX* proteins.

## Results

### Alternative *CaaX* motifs negatively impact Ydj1p-dependent thermotolerance

The yeast Hsp40 chaperone Ydj1p is a farnesylated protein. Ydj1p is required for growth at elevated temperatures, and its farnesylation status influences this phenotype (*Caplan et al., 1992*). Using a qualitative plate growth assay, we observed as previously reported that yeast fail to grow at elevated temperatures in the absence of Ydj1p (*Figure 1A*; vector). Expression of wildtype Ydj1p (CASQ) reversed this phenotype and allowed growth at elevated temperature. The non-prenylatable Ydj1p mutant (SASQ) failed to grow at the highest temperature assessed (40°C) and displayed reduced growth at a slightly lower temperature (37°C) as evident by the smaller colony size relative to wildtype Ydj1p. We decided to take advantage of the *ydj1Δ* temperature sensitivity phenotype to develop Ydj1p as a novel reporter for post-isoprenylation processing events.

Our initial goal was to determine whether Rce1p and/or Ste24p-mediated cleavage of the Ydj1p *CaaX* motif influenced thermotolerance. At the onset of this study, we expected that altering the *CaaX* motif of Ydj1p would yield phenotypes matching that of either farnesylated or unmodified Ydj1p (*i.e.* CASQ and SASQ phenotypes respectively). We further expected farnesylated forms of Ydj1p to be cleaved and carboxylmethylated at the COOH-terminus. We observed an intermediate phenotype, however, when we evaluated Ydj1p mutants containing the *CaaX* motifs from **a**-factor (CVIA) and the Ste18p Gγ subunit (CTLM). These Ydj1p *CaaX* mutants developed normal colony size at 37°C, indicative of isoprenylation, but grew poorly at 40°C relative to wildtype Ydj1p. The growth defect at 40°C was more pronounced than just a smaller colony size. There were fewer colonies per

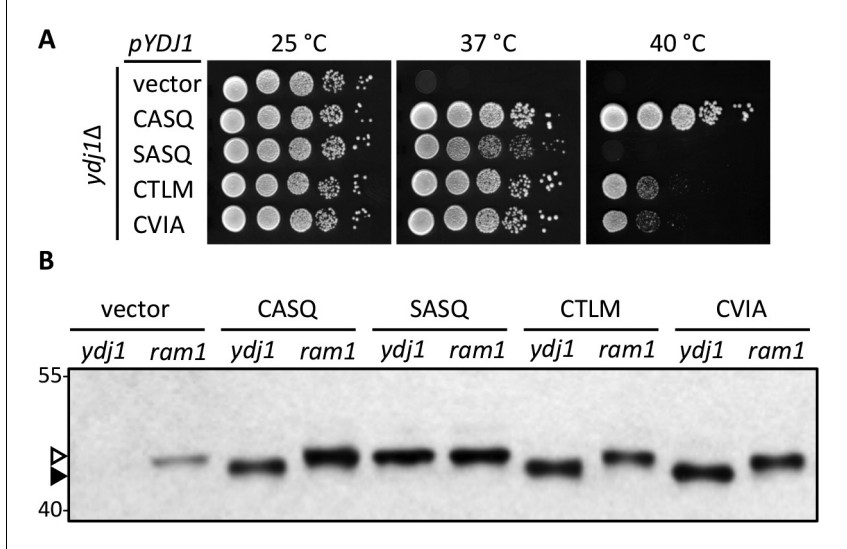

**Figure 1.** Alternate CaaX motifs on Ydj1p affect thermotolerance but not farnesylation. (**A**) Yeast cultured in selective SC-Ura media were normalized for culture density and spotted as 10-fold serial dilutions onto YPD; the leftmost spot in each panel is undiluted. Plates were incubated at the indicated temperature as described in *Materials and Methods*. The strain used was yWS304 (*ydj1Δ*); the *CEN* plasmids used were pRS316, pWS942, pWS1132, pWS1246, and pWS1286. Similar results were observed for Ydj1p *CaaX* variants expressed at the chromosomal level (see *Figure 1—figure supplement 1*). (**B**) Lysates from the indicated genetic backgrounds were prepared from cultures grown at 25°C in SC-Ura and analyzed by immunoblot with Ydj1p antiserum. The strains used were yWS304 (*ydj1Δ*) and yWS1632 (*ram1Δ*); the *CEN* plasmids used were the same as reported for panel A. The *ram1* strain lacks farnesyl transferase activity and produces unmodified Ydj1p (open triangle) having slower mobility than farnesylated Ydj1p (closed triangle). Approximately 25% of the signal in the *ram1* samples comes from the chromosomal copy of Ydj1p present in that strain (see *Figure 1—figure supplement 2*). Values to the left of the image indicate the migration of protein standards (kDa). The farnesylation of the *CaaX* variants is not temperature dependent (see *Figure 1—figure supplement 3*). The data in panel A are representative of multiple biological replicates (n= 2, full set of strains; n=7, all but vector strain). The data in panel B are representative of 2 biological replicates.

The following figure supplements are available for figure 1:

**Figure supplement 1.** Thermotolerance profiles of chromosome-encoded Ydj1p *CaaX* mutants.

**Figure supplement 2.** Comparison of genomic and plasmid-based expression of Ydj1p in the ram1 strain.

**Figure supplement 3.** Isoprenylation status of Ydj1p CaaX variants at high temperature.

spot relative to farnesylated Ydj1p (CASQ) and additional colonies did not appear after prolonged growth at elevated temperature. It should be mentioned that the *CaaX* motifs compared were purposefully chosen. In the context of **a**-factor, the CASQ motif is recognized as being Ste24p-specific, CTLM is Rce1p-specific, and CVIA is cleaved by both proteases (*Trueblood et al., 2000*); this target specificity is species-independent (*Plummer et al., 2006*; *Cadiñanos et al., 2003*).

Initial experiments were performed using Ydj1p *CaaX* variants encoded on low-copy plasmids. To insure that mild over-expression was not overwhelming the activities of CaaX-processing enzymes and thus confounding phenotypes and data interpretation, we evaluated Yd1jp mutants in the context of chromosome encoded genes. Nearly identical results were obtained (*Figure 1—figure supplement 1*).

## Alternative *CaaX* motifs do not hinder Ydj1p farnesylation

We predicted that the differential phenotype observed for Ydj1p *CaaX* mutants (SASQ, CTLM and CVIA) was due to differential expression and/or post-translational modification relative to wildtype

(CASQ). For SASQ, we expected loss of isoprenylation. For the others, we expected either reduced farnesylation or alternative geranylgeranylation. To assess the isoprenylation status of the Ydj1p mutants, we took advantage of the observation that farnesylation increases the SDS-PAGE mobility of Ydj1p (*Caplan et al., 1992*). Lysates from the same strains used for the thermotolerance assay were evaluated by immunoblot, which revealed that SASQ was indeed unmodified while CTLM and CVIA were fully farnesylated (*Figure 1B*). Moreover, the expression levels of the *CaaX* variants appeared similar. The *ram1* genetic background was used to generate unmodified Ydj1p for comparison; *RAM1* encodes the β subunit of the farnesyl isoprenyltransferase. The *ram1* strain contains a chromosomal copy of *YDJ1* that contributes a small amount of immunoreactive signal in each of the *ram1* samples; the contribution was calculated to be approximately 25% of the total signal in plasmid-transformed strains though comparison of serially diluted samples (*Figure 1—figure supplement 2*). It is important to note that the cultures for the extracts evaluated by immunoblotting were prepared at permissive temperature (25°C). Thus, it remains formally possible that reduced thermotolerance could result from poor farnesylation of CTLM and CVIA sequences at higher temperatures. We find this unlikely based on evidence that these motifs are still farnesylated at higher temperature when using a genetic background that allows growth at 40°C (see below and *Figure 1—figure supplement 3*).

## Post-isoprenylation events decrease the thermotolerance and growth of strains expressing Ydj1p *CaaX* variants

The differential phenotypes observed for Ydj1p (CASQ) and the Ydj1p mutants (CTLM and CVIA) suggested that the modifications occurring to wildtype Ydj1p were somehow different than those occurring to the mutants. In the absence of altered isoprenylation as an explanation for the differing phenotypes, we considered that post-isoprenylation events might differ between the variants even though we initially predicted that each Ydj1p variant would be proteolytically trimmed to the same species. In keeping with the reported target specificity of the *CaaX* proteases, we expected that Ste24p would cleave Ydj1p (CASQ), Rce1p would cleave Ydj1p (CTLM), and both proteases would act on Ydj1p (CVIA). Because *CaaX* proteolysis and carboxylmethylation are coupled events, we also expected each species to be carboxylmethylated by Ste14p.

Using the qualitative plate growth assay, we evaluated the impact of expressing Ydj1p *CaaX* variants on thermotolerance in the absence of Ste24p (*ste24Δ ydj1Δ*), Rce1p (*rce1Δ ydj1Δ*), or Ste14p (*ste14Δ ydj1Δ*) (*Figure 2*). No change in patterns was observed in the absence of Ste24p (*i.e.* compare to *Figure 1A,* 40°C). Lack of Rce1p improved the thermotolerance of the strain expressing Ydj1p (CTLM) but no other strain, including Ydj1p (CVIA). Lack of Ste14p improved thermotolerance for strains expressing Ydj1p (CTLM) or Ydj1p (CVIA). A similar pattern was observed using chromosome-encoded Ydj1p *CaaX* variants (*Figure 1—figure supplement 1*). We also attempted to examine thermotolerance in a genetic background devoid of *CaaX* protease activity (*i.e. ydj1Δ rce1Δ ste24Δ*). A genetic cross was used to create a parent diploid (*YDJ1/ydj1Δ RCE1/rce1Δ STE24/ste24Δ*), but isolation of the desired haploid strain through sporulation was hampered by a spore germination defect for the triple mutant (Hildebrandt and Schmidt, unpublished observation). Ultimately, germination was achieved by including a plasmid copy of *YDJ1* in the parent strain. Once the triple knockout haploid strain was recovered and the plasmid lost, the strain did not grow well, even at room temperature, precluding its use in our study.

In addition to the thermotolerance phenotypes observed, we noticed that over-expression of Ydj1p had a negative impact on cell growth, even for wildtype Ydj1p (CASQ) expressed in a wildtype background (*YDJ1*). This negative effect was initially observed in colonies surviving selection after transformation and was a persistent phenotype upon colony purification (Hildebrandt and Schmidt, unpublished observation). We took advantage of this observation to examine the impact of alternative Ydj1p *CaaX* motifs on this phenotype. By comparison to a vector control, strains over-expressing Ydj1p (CASQ) or non-prenylated Ydj1p (SASQ) yielded slightly smaller colonies after a fixed time of incubation on selective solid media and a 2-fold increase in liquid culture doubling time in both the presence and absence of *STE14* (*Figure 3*; compare vector with CASQ and SASQ). These observations suggest that farnesylation and post-isoprenylation modification are not a contributing factor to the negative impact of over-expressed Ydj1p. When strains over-expressing Ydj1p *CaaX* variants (CTLM and CVIA) were evaluated, a more pronounced colony growth defect and significantly increased doubling times were observed in the presence of *STE14*. The phenotypes associated with

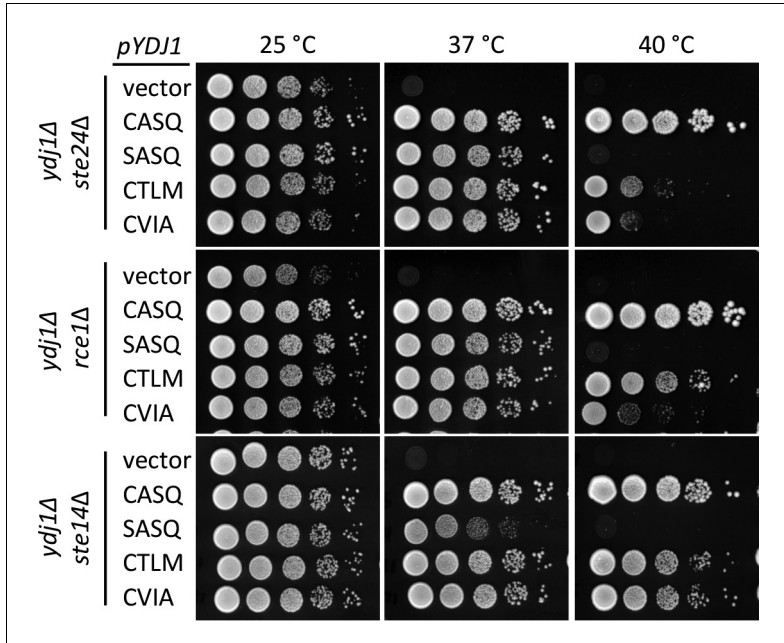

**Figure 2.** Post-isoprenylation processing negatively impacts Ydj1p variants having alternative CaaX motifs. Data were collected as described in **Figure 1**. The strains used were yWS1693 (*ydj1Δ ste24Δ*), yWS1689 (*ydj1Δ rce1Δ*), and yWS1635 (*ydj1Δ ste14Δ*); the *CEN* plasmids used are the same as described in **Figure 1** except that the vector used for yWS1693 and yWS1689 was pRS416. The data for each genetic background are representative of 2 biological replicates.

the Ydj1p *CaaX* variants were suppressed in the absence of *STE14* such that growth rates approached those observed for Ydj1p (CASQ).

## Alternative *CaaX* motifs disrupt Ydj1p localization

Post-isoprenylation events modulate the localization of the Ras GTPase and other *CaaX* proteins (**Boyartchuk et al., 1997**; **Roberts et al., 2008**; **Bergo et al., 2000**; **Bergo et al., 2002**). We thus evaluated the impact of alternative *CaaX* motifs on the localization of Ydj1p. Previous studies have reported that Ydj1p is primarily cytosolic upon differential fractionation, but has a minor association with membranes presumed to be ER and mitochondria (**Caplan and Douglas, 1991**). Using a low copy GFP-Ydj1p reporter system, we observed altered distribution of Ydj1p *CaaX* variants relative to wildtype Ydj1p (**Figure 4A**). Strains expressing Ydj1p *CaaX* variants (CTLM and CVIA) commonly displayed puncta that were not generally observed in yeast expressing wildtype Ydj1p or non-prenylated Ydj1p (SASQ). Quantification revealed that puncta was the major phenotype in Ydj1p *CaaX* variants (CTLM and CVIA) and non-existent in cells expressing wildtype Ydj1p or non-prenylated Ydj1p (SASQ) (**Figure 4B**). The normal distribution for the Ydj1p *CaaX* variants (CTLM and CVIA) was largely restored in a strain lacking Ste14p (*ydj1Δ ste14Δ*). The above results were obtained in a genetic background lacking Ydj1p (*i.e ydj1Δ*); similar results were observed when Ydj1p *CaaX* variants were expressed in a genetic background wildtype for Ydj1p (**Figure 4—figure supplement 1A**). Collectively, our observations suggest that the post-translational processing of Ydj1p variants (CTLM and CVIA) results in an altered subcellular distribution of Ydj1p.

To assess the functionality of GFP-tagged Ydj1p *CaaX* variants, we compared their thermotolerance to that of untagged counterparts. At 37°C, we observed that non-prenylated GFP-Ydj1p (SASQ) was less thermotolerant than it's untagged counterpart (**Figure 4C**). This effect was again observed at 40°C where all the GFP-tagged versions failed to grow, including GFP-Ydj1p (CASQ). Farnesylation of the GFP-Ydj1p *CaaX*-variants was not impaired, however (**Figure 4—figure supplement 1B**). These results indicate that that the GFP-fusion attenuates Ydj1p function to some extent. Despite the compromised function, the trend observed for the GFP-variants was similar to their

**Figure 3.** Over-expression of Ydj1p CaaX variants results in STE14 dependent slow growth. Yeast cultured in selective SC-Ura,Leu media were normalized for culture density and spotted as 10-fold serial dilutions onto selective SC-Ura,Leu; the leftmost spot in each panel is undiluted. Plates were incubated at 30°C. Growth rates of the strains were determined in SC-Ura,Leu liquid media at 30°C. The strain used was SM1188 (*ste14Δ*), which was transformed with two plasmids. The plasmids used for over-expression of Ydj1p were pWS948, pWS972, pWS1247, and pWS1291; pSM703 was the representative empty vector. The plasmid used for expression of Ste14p was pSM1316; pRS315 was used for the vector (-) condition. The serial dilution data are representative of 2 biological replicates. The doubling times are averages of 4 biological replicates; error ranges represent the 95% confidence interval (see *Figure 3—source data 1*).

The following source data is available for figure 3:

**Source data 1.** Spreadsheet with raw values for $A_{600}$ vs. time of yeast over-expressing Ydj1p *CaaX* variants in the presence or absence of *STE14*.

untagged counterparts (i.e. non-prenylated GFP-Ydj1p was more thermosensitive than the GFP-Ydj1p *CaaX* variants).

## Biophysical analysis of the Ydj1p COOH-terminus

A model that fits our data is one where the post-isoprenylation processing differs between Ydj1p (CASQ) and Ydj1p *CaaX* variants (CTLM and CVIA). In particular, the latter are modified in such a way that results in altered phenotypes – thermotolerance and localization. This model predicts, contrary to initial expectations, that Ydj1p (CASQ) is uncleaved and thus not carboxylmethylated after initial isoprenylation. By contrast, Ydj1p *CaaX* variants (CTLM and CVIA) are fully processed. Consistent with the reported target specificities of the *CaaX* proteases, Ydj1p (CTLM) would be cleaved by Rce1p, and Ydj1p (CVIA) would be cleaved by either Rce1p or Ste24p.

To test our model directly, we used mass spectrometry to analyze the modifications associated with purified Ydj1p (CASQ) (*Figure 5* and *Figure 5—figure supplement 1*). This analysis yielded 65% coverage of the protein, including a COOH-terminal peptide. The COOH fragment was farnesylated and retained the CASQ *CaaX* motif. Peptides that were unfarnesylated, or farnesylated and partially or fully modified at the COOH-terminus were specifically searched for but not detected. This observation provides direct support for our working model.

## The limited processing of the CASQ motif is not reporter dependent

The alternative *CaaX* motifs evaluated in the context of Ydj1p were initially chosen to represent the motifs that could be recognized by the Rce1p and Ste24p *CaaX* proteases. In the context of the **a-**

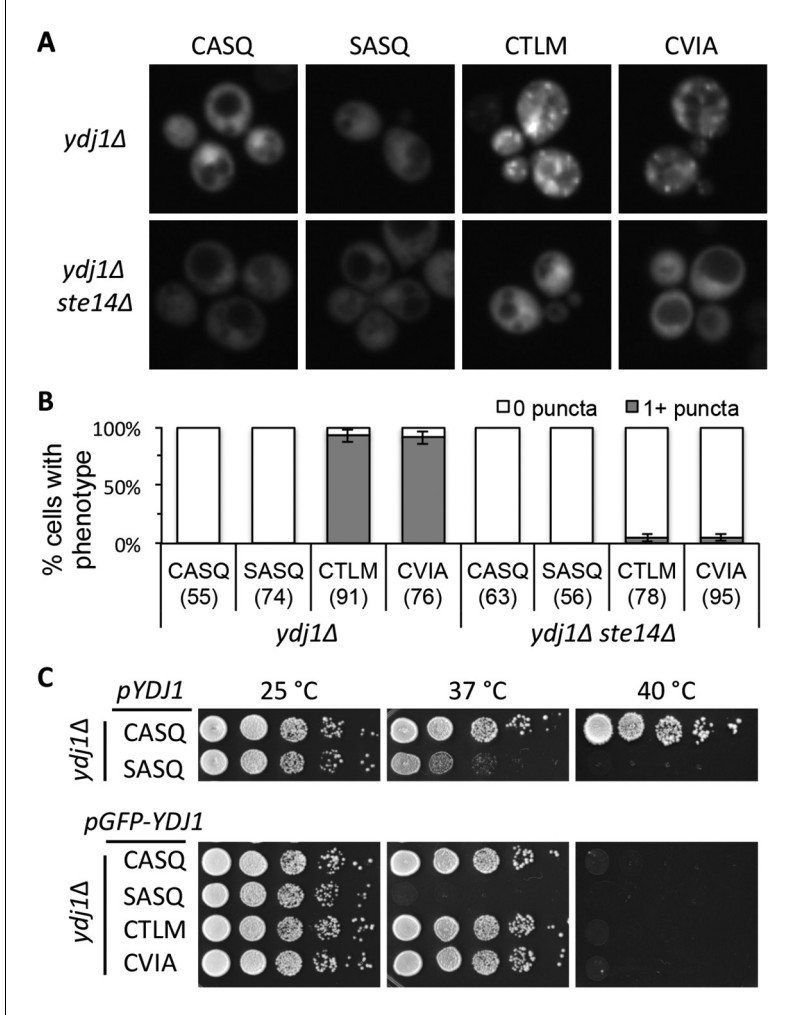

**Figure 4.** Ydj1p variants with alternative *CaaX* motifs are mislocalized. (**A**) The indicated GFP-Ydj1p *CaaX* variants were expressed in yWS304 (*ydj1Δ*) or yWS1635 (*ydj1Δ ste14Δ*). Images of multiple fields were collected under fluorescence optics, with images representative of the majority phenotype being shown. The *CEN* plasmids used were pWS1389-1392, encoding *CaaX* variants CASQ, SASQ, CTLM and CVIA, respectively. (**B**) Quantification of phenotypes observed in panel A. Values in parentheses associated with each motif indicate the total number of cells classified over the course of two independent experiments. Bars indicate the average number of cells observed to have the indicated phenotype. (**C**) Thermotolerance of strains expressing GFP-Ydj1p *CaaX* variants compared to untagged versions. The differences are not due to farnesylation defects (see *Figure 4—figure supplement 1*). Data were collected as described in *Figure 1*. The strain used was yWS304 (*ydj1Δ*); the *CEN* plasmids used were pWS942, pWS1132, and pWS1389-1392. Data in panel A are representative of 4 biological replicates; data in panel B were calculated from 2 biological replicates with error bars representing the range observed between replicates (see *Figure 4—source data 1*); data in panel C are representative of 3 biological replicates.

The following source data and figure supplement are available for figure 4:

**Source data 1.** Spreadsheet with raw values used for percent calculations of puncta in GFP-Ydj1p *CaaX* variant strains.

**Figure supplement 1.** Localization and farnesylation of GFP-Ydj1p variants with alternative *CaaX* motifs.

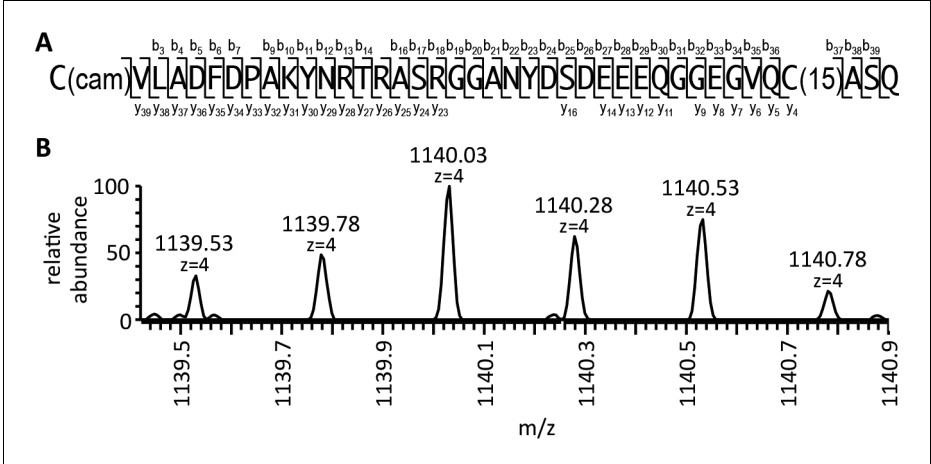

**Figure 5.** Biophysical analysis of C-terminal peptide derived from Ydj1p. (**A**) His$_6$-Yd1jp was expressed in yWS304 (*ydj1Δ*), isolated by immobilized metal affinity chromatography, and analyzed by mass spectrometry after digestion with endoproteinase GluC. The plasmid used was pWS1307. The sequence indicates the detected b$^-$ and y$^-$ fragments of the C-terminal peptide observed with MS/MS (see *Figure 5—figure supplement 1*). C(cam) is carbamidomethyl cysteine; C(15) is isoprenylated cysteine. (**B**) Full MS of the detected C-terminal peptide, with m/z value and mass accuracy of the species indicated. The same result was observed in two independent His$_6$-Ydj1p samples.

The following figure supplement is available for figure 5:

**Figure supplement 1.** MS/MS spectrum of the farnesylated C-terminal peptide.

factor reporter, the CVIA naturally present on **a**-factor is cleaved by both proteases, whereas CTLM is Rce1p-specific, and CASQ is Ste24p-specific (*Trueblood et al., 2000*; *Plummer et al., 2006*; *Cadiñanos et al., 2003*). Yet, our findings indicate that the CASQ motif is not cleaved in the context of Ydj1p. To reconcile these observations, we considered that cleavage of CASQ was either context-specific or an over-interpretation of **a**-factor-based results. The latter seemed more likely given that previous studies, including our own, typically used over-expression systems and/or highly sensitized methods for detection of **a**-factor bioactivity (*Trueblood et al., 2000*; *Plummer et al., 2006*; *Cadiñanos et al., 2003*; *Marcus et al., 1991*; *Huyer et al., 2006*). Thus, we decided to re-investigate the impact of the CASQ motif on **a**-factor production using conditions that minimize over-expression effects. The system was used in conjunction with both qualitative and quantitative mating assays to assess **a**-factor bioactivity across a wide range of production levels.

Our initial analysis of **a**-factor production confirmed previously reported findings of protease specificity for the CVIA, CASQ, and CTLM motifs (*Figure 6A*). In this genetic test, the formation of mating products (*i.e.* diploid colonies) is directly related to **a**-factor production, which must be farnesylated, cleaved, and carboxylmethylated in order to be secreted and function as a mating signal. Using similar inputs of *MAT***a** cells capable of producing **a**-factor, the pattern of diploid formation was similar in the context of the natural **a**-factor sequence (CVIA) whether Rce1p or Ste24p was used to drive pheromone production. A similar pattern was observed for Rce1p-dependent mating in the context of CTLM. Fewer diploid colonies, however, appeared for Ste24p-dependent mating in the context of CASQ, indicative of reduced **a**-factor production. Because the protease and **a**-factor genes (*RCE1*, *STE24*, *MFA1*) were slightly over-expressed through the use of *CEN*-based plasmids in this experiment, we re-evaluated **a**-factor production in the context of the chromosomal genes for the proteases (*Figure 6B*). This analysis revealed no differences in diploid production for the CVIA and Rce1-dependent CTLM conditions, but much fewer diploids were evident for the Ste24p-dependent CASQ condition. An analysis of mating efficiencies for these strains revealed that **a**-factor (CASQ) was approximately 1% as effective as **a**-factor (CVIA) and the CTLM variant. Since the **a**-factor gene was encoded on a *CEN*-based plasmid, we expect that mating efficiency observed for **a**-

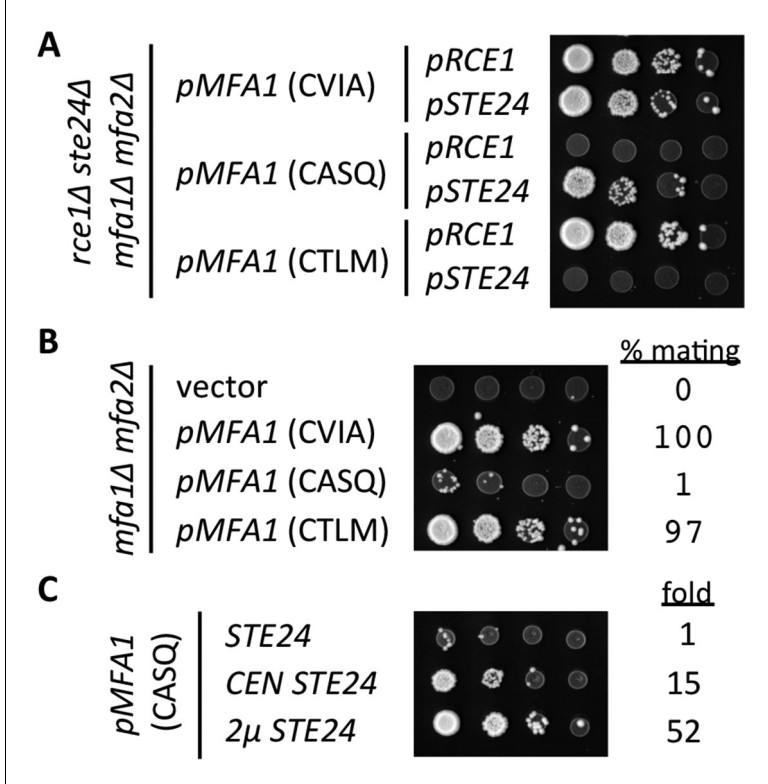

**Figure 6.** Impact of different CaaX motifs on **a**-factor bioactivity. (**A**) *MAT***a** yeast co-expressing the indicated **a**-factor species and *CaaX* protease were normalized for cell density then serially diluted in the presence of 10-fold or higher excess *MAT***a** cells. The mixtures were spotted onto media selective for diploid growth. Plasmid-derived strains of yWS164 (*rce1Δ ste24Δ mfa1Δ mfa2Δ*) were created using appropriate combinations of *CEN* plasmids pSM1275, pSM1093, pWS610, and pWS612-13. (**B**) Mating mixtures were prepared and analyzed as described for panel A. Equal portions of each mating mixture were also analyzed for mating efficiency as described in the *Materials and Methods* section, where the condition involving wildtype *MFA1* was set to 100%. Plasmid-derived strains of SM2331 (*mfa1Δ mfa2Δ*) were created using pRS415 and the **a**-factor encoding plasmids described for panel A. (**C**) Mating mixtures were prepared and analyzed as described for panel B. The value for fold refers to the relative mating efficiency observed for each condition, where the condition involving chromosome-encoded *STE24* was set as the reference. Plasmid-derived strains were created by transformation of SM3689 (*rce1Δ mfa1Δ mfa2Δ*) and yWS164 with pWS612 and either pRS316, pSM1093, or pSM1194. The data in panel A are representative of 3 biological replicates; the serial dilutions in panels B and C are representative of 2 technical replicates with reported values derived from the averages of 2 technical replicates. Transfer of the CASQ motif to Ras2p also results in an 'uncleaved' phenotype (see *Figure 6—figure supplement 1*).

The following figure supplement is available for figure 6:

**Figure supplement 1.** Impact of different *CaaX* motifs on GFP-Ras2p farnesylation and localization.

factor (CASQ) is still an over-estimate. Nevertheless, the data clearly indicate that the CASQ motif is associated with significantly decreased **a**-factor bioactivity.

Our observations suggested that at least one post-translational modification is inefficient for **a**-factor CASQ (*i.e.* reduced farnesylation, proteolysis or carboxylmethylation). To differentiate these possibilities, we examined the impact of *STE24* gene dosage on mating that was dependent on **a**-factor (CASQ). This analysis revealed that increasing gene dosage yielded increasingly better mating efficiencies (*Figure 6C*). This result implies that cleavage is the rate-limiting step for pheromone production from **a**-factor (CASQ), rather than reduced farnesylation or carboxylmethylation. This result also indicates that the inefficient cleavage of CASQ is transferrable to different reporters and thus an intrinsic property of the motif itself.

We also tested the impact of CASQ on a Ras-based reporter. In yeast, the localization of Ras2p (CIIS) to the plasma membrane is highly dependent on the canonical *CaaX* modification pathway (*i.e.* farnesylation, proteolysis, and carboxylmethylation) (*Boyartchuk et al., 1997*; *He et al., 1991*; *Manandhar et al., 2010*). In fact, defective proteolysis alone is sufficient to cause mislocalization. Using GFP-Ras2p, we observed that the CASQ variant was farnesylated but not properly localized to the plasma membrane (*Figure 6—figure supplement 1*). These findings provide yet additional support that the ability of CASQ to be farnesylated and resist cleavage are reporter-independent properties of the motif.

## Interactions between Ydj1p and the Rnq1 prion domain are not affected by *CaaX* motif mutations

It is unclear why Ydj1p *CaaX* mutants display growth defects (i.e. thermosensitivity, over-expression toxicity). One possibility is that client interactions are disrupted. This could be due to a previously unrecognized role for the *CaaX* motif in physical interaction and/or a reduction in the availability of Ydj1p due to its mislocalization (*e.g.* CTLM and CVIA). One well-described interactor is the Gln/Asn-rich prion domain of Rnq1p (PrD); the prion form of Rnq1p is referred to as [$RNQ^+$] or [$PIN^+$]. Ydj1p suppresses PrD proteotoxicity in a farnesyl-dependent manner (*Summers et al., 2009*). Genetically, Ydj1p *CaaX* mutants (CTLM and CVIA) yield a phenotype similar to wildtype Ydj1p in suppressing PrD-dependent proteotoxicity, whereas unfarnesylated Ydj1p (CASQ) displayed poor growth (*Figure 7A*). Biochemically, co-immunoprecipitation analysis (coIP) revealed interaction between Prd and farnesylated forms of Ydj1p (CASQ, CTLM, and CVIA), whereas unfarnesylated Ydj1p (SASQ) failed to interact (*Figure 7B*). We also investigated the Ydj1p-dependent regulation of Axl1p levels and interaction between Ydj1p and the polyglutamine-rich fragment of Huntingtin (Htt) (*Meacham et al., 1999*; *Muchowski et al., 2000*; *Meriin et al., 2002*). These studies also did not reveal functional differences between wildtype Ydj1p and farnesylated *CaaX* mutants (CTLM and CVIA).

## Discussion

We have provided evidence that the post-translational processing of yeast Ydj1p deviates from the canonical model observed for *CaaX* proteins, particularly the Ras GTPases and the **a**-factor mating pheromone (*Figure 8A*). Unlike Ras and **a**-factor that are isoprenylated, proteolyzed and carboxylmethylated, Ydj1p is shunted out of the pathway such that it retains its *CaaX* motif after isoprenylation. When the complete set of *CaaX* modifications is applied to Ydj1p, cellular physiology is altered in a way that manifests several distinct phenotypes - altered thermotolerance, growth rates, and subcellular localization. The specific underlying cellular process that affects thermotolerance and growth rates is unknown, but could stem from any one of the many processes with which Ydj1p has been associated. Our investigations of client interactions did not yield differences for wildtype and mutant Ydj1p. Additional studies will be needed to establish whether retention of the *CaaX* motif is important for certain protein-protein interactions, proper localization, or for optimal biochemical activity of Ydj1p. We cannot discern between these possibilities from available data.

Collectively, our observations imply that shunting is required to prevent carboxylmethylation of Ydj1p rather than *CaaX* proteolysis per se. We conclude this because disrupting Ste14p reverses the altered phenotypes associated with Ydj1p variants having cleavable *CaaX* motifs. If preventing *CaaX* proteolysis were the main goal of the shunt motif, then carboxylmethylation would have had little or no impact. Because *CaaX* proteolysis precedes carboxylmethylation, regulating proteolysis through use of different *CaaX* sequences is thus an easy way for the cell to control carboxylmethylation of target proteins. Carboxylmethylation is generally described as enhancing the non-polar/hydrophobic character of the COOH-terminus. In cases where carboxylmethylation occurs (e.g. **a**-factor, Ras-related GTPases, and prelaminA), the effect appears to enhance membrane partitioning of the molecule (*Bergo et al., 2000*; *Marcus et al., 1991*; *Michaelson et al., 2005*; *Takahashi et al., 2005*; *Hanker et al., 2010*; *Ibrahim et al., 2013*). Ydj1p, by comparison, is widely regarded as being mostly a cytosolic chaperone despite being farnesylated. Future studies will address whether maintenance of a polar COOH-terminus (i.e. shunting) is critical for modulating Ydj1p interactions with membranes.

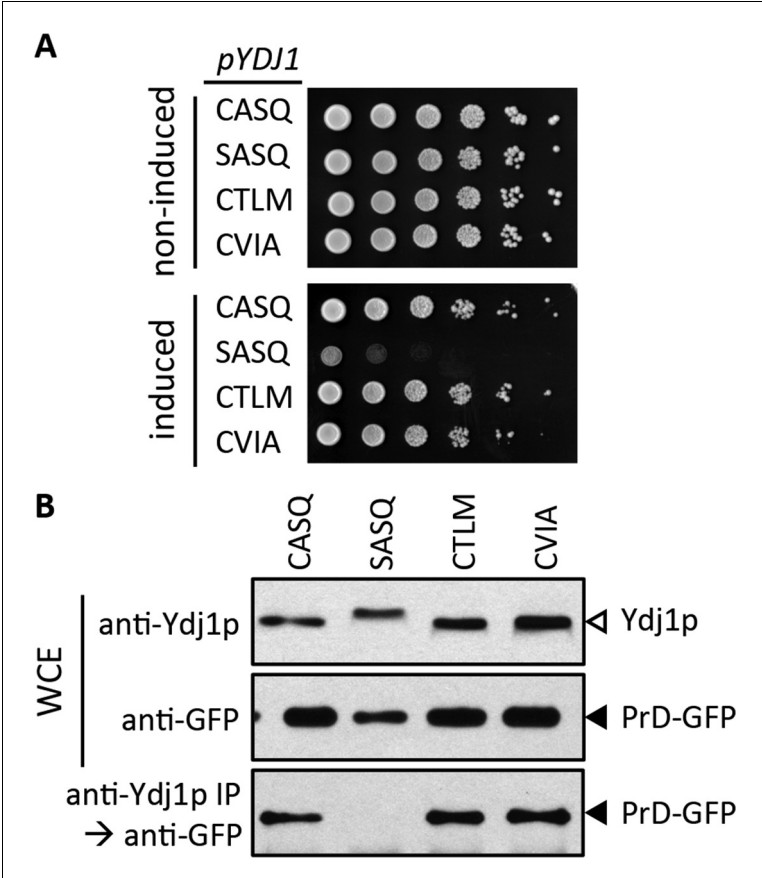

**Figure 7.** Impact of different *CaaX* motifs on Ydj1p client interactions. (**A**) Yeast containing an inducible PrD expression plasmid and the indicated Ydj1p *CaaX* variants were spotted as a serial dilution series on media containing glucose (non-induced) or galactose (induced) and incubated at 30℃. The strains used are plasmid derivatives of yWS2078 ([*RNQ⁺*] *ydj1Δ*) created using pWS1430 (PrD) and pWS1326-1329 (Ydj1p). (**B**) Yeast co-expressing PrD-GFP and the indicated Ydj1p *CaaX* variants were evaluated for interactions by coIP. Whole cell extracts (WCE) were prepared, and 10 µg of each lysate was analyzed by SDS-PAGE and immunoblot using the indicated antibody, while 100 µg of each lysate was immunoprecipitated using Ydj1p antibody and subsequently evaluated for recovery of PrD-GFP with GFP antibody. The strains used are similar to those described for panel A except that pWS1431 (PrD-GFP) was used as the source of PrD.

A major question that our observations raise relates to the frequency of shunting among *CaaX* proteins. There are certainly other *CaaX* proteins that are strong candidates for utilizing the shunt pathway. These isoprenylated proteins have been characterized to retain their *CaaX* motifs directly through biophysical methods (*i.e.* Phk α1 and β subunits; Gγ5) or indirectly through an inability to detect carboxylmethylation (*i.e.* Rab38) (*Leung et al., 2007*; *Heilmeyer et al., 1992*; *Kilpatrick and Hildebrandt, 2007*). Our study adds Ydj1p the list of shunted proteins, but importantly, it is the first to associate a negative impact on cellular processes when a shunted protein is forced to undergo post-isoprenylation processing (i.e. proteolysis and carboxylmethylation). This implies, for Ydj1p at least, that a retained *CaaX* motif is a critical feature necessary for optimal function of the protein. There is a high degree of sequence conservation within the *CaaX* motif for the DNAJA2 chaperone family, to which Ydj1p belongs (see *Figure 8B*). This suggests that our observations may not be specific to yeast. It remains to be determined whether any negative consequences arise when other shunted proteins are forcibly modified.

It is reasonable to assume that the amino acid sequences that direct a protein to the shunt pathway must be fit two rules basic rules: (1) recognition by prenyltransferases, and (2) not recognized by *CaaX* proteases. The amino acid rules of the *CaaX* motif that promote shunting, however, remain

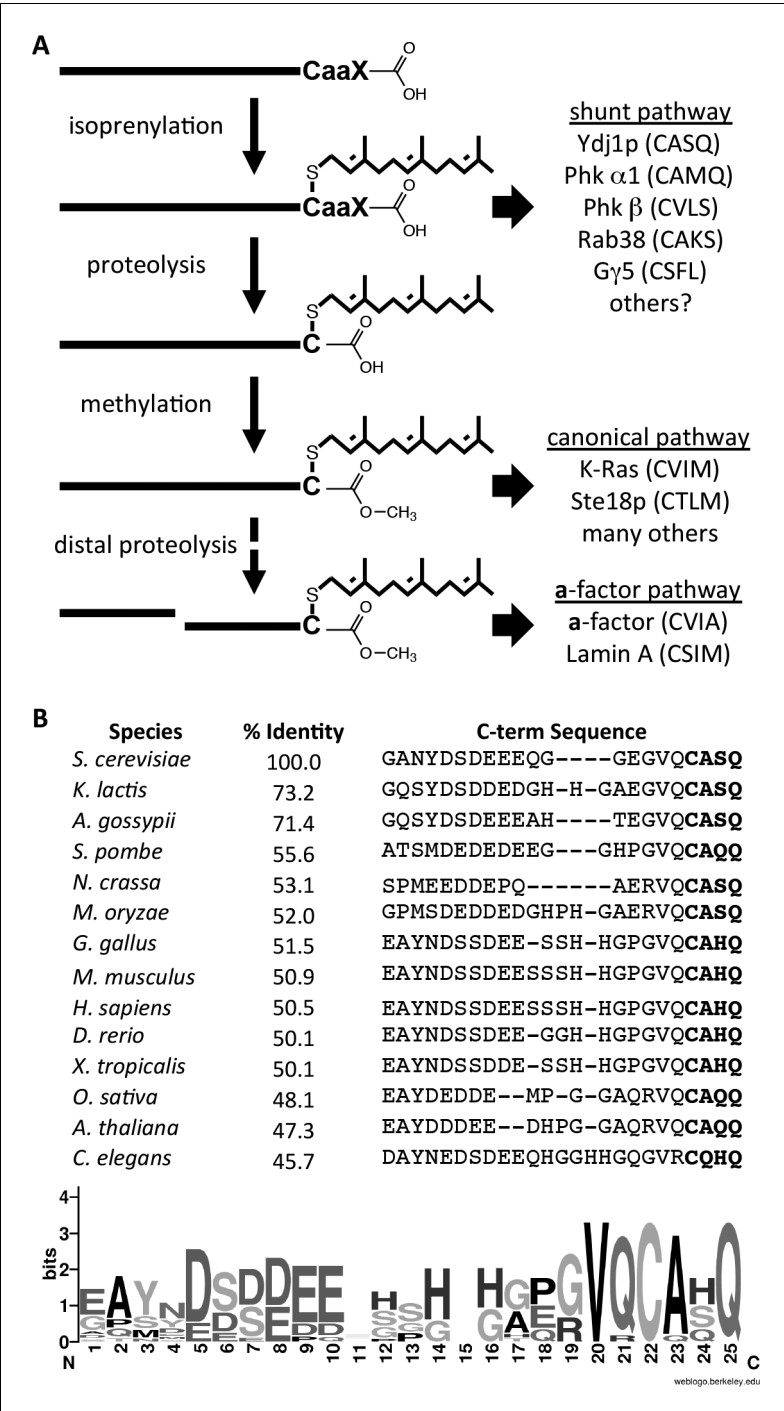

**Figure 8.** Model for post-translational modification of Ydj1p. (**A**) The *CaaX* motif directs isoprenylation of the protein. The isoprenylated species is either the endpoint modification (e.g. Ydj1p) or an intermediate for further modification (e.g. Ras, a-factor, lamin A). (**B**) Alignment of the COOH-termini of Ydj1p and its homologs along with percent identity scores for entire sequences relative to *S. cerevisiae* Ydj1p. The sequences and alignment scores were retrieved from the Homologene database (http://www.ncbi.nlm.nih.gov/homologene). A WebLogo representation of amino acid frequency within the COOH-terminal region is shown.

unclear. Limited alanine-scanning site-directed mutagenesis of Ydj1p has not revealed a single amino acid within or immediately adjacent to the motif that is critical for shunting (Kim and Schmidt, unpublished observations). By comparison, similar studies of the shunted protein Gγ5 (CSFL) suggest that the presence of an aromatic residue at the $a_1$ or $a_2$ position of the *CaaX* motif is the feature that reduces cleavage efficiency (*Kilpatrick and Hildebrandt, 2007*). Ydj1p and Phk subunits, however, lack an aromatic residue at this position. Moreover, we have observed that the CASQ and CTSQ *CaaX* motifs support similar phenotypes in the context of both Ydj1p and **a**-factor (*i.e.* poorly cleaved), implying that the $a_1$ position can be something other than aliphatic in nature ([*Plummer et al., 2006*]; Hildebrandt and Schmidt, unpublished observation). Thus, the feature(s) that dictates whether a *CaaX* protein is shunted is likely more complex than a specific amino acid type at a particular position.

The CASQ sequence taken advantage of in this study is the best example to date of a *CaaX* motif that is specifically cleaved by Ste24p (*Trueblood et al., 2000*). Our data continue to support this specificity, but only in the context of **a**-factor as a reporter and only under conditions of protease and/or substrate over-expression. By our best methods, we have determined that only a small amount of fully processed **a**-factor is generated with the CASQ motif relative to motifs cleaved by Rce1p (*e.g.* CVIA, CTLM). Likewise, it is possible that a very small amount of Ydj1p (CASQ) is cleaved, but not in sufficient amounts to be detected. Thus, we conclude that CASQ should not be technically referred to as a Ste24p-specific *CaaX* motif. In fact, emerging evidence points to roles for Ste24p in activities that do not appear to involve isoprenylated targets. These range from quality control of protein topology and translocation to cleavage of non-prenylated peptides (*Ast et al., 2016*; *Hildebrandt et al., 2016*; *Tipper and Harley, 2002*). Taken together, the majority of evidence suggests that Ste24p should be re-classified as a promiscuous protease rather than a highly specific and efficient *CaaX* protease.

Through our development of Ydj1p as a novel CaaX protein reporter, we have been able to make two significant observations. First, a shunt pathway exists to divert *CaaX* proteins such that isoprenylation is the only post-translational event occurring at the *CaaX* sequence. This pathway is critical to maintenance of Ydj1p dependent phenotypes and likely critical to other proteins that use the shunt pathway. Second, we have undercut the argument that CASQ is a Ste24p-specific *CaaX* motif, which leaves open the question of whether such motifs exist. These observations add significant complexity to protein prenylation, its regulation, and the impact of inhibitors that target the various steps associated with protein prenylation.

# Materials and methods

## Yeast strains

The yeast strains used in this study are listed in *Table 1*. Strains were isolated from a commercial *MAT***a** haploid genomic deletion library ([*Shoemaker et al., 1996*]; currently available from transO-MIC Technologies Inc., Huntsville, AL), obtained from the yeast community, or created by standard yeast methods. Strains were routinely propagated at 30°C, or room temperature if temperature sensitive, on either YPD or selective media as appropriate. Plasmids were introduced into strains via a lithium acetate-based transformation procedure (*Elble, 1992*).

yWS1635 (*MAT***a** *his3Δ1 leu2Δ0 met15Δ0 ura3Δ0 ydj1::KAN^R ste14::KAN^R*) was derived by genetic cross between yWS1577 (*MATα his3Δ1 leu2Δ0 met15Δ0 ura3Δ0 ydj1::KAN^R*) and yWS1626 (*MAT***a** *his3Δ1 leu2Δ0 met15Δ0 ura3Δ0 ste14::KAN^R*) followed by random sporulation and evaluation of germinated spores by PCR and appropriate genetic phenotypes to identify haploids with the appropriate genotype. yWS1577 is a *MATα* derivative of yWS304 that was created using an inducible plasmid-based mating type-switching system ($P_{GAL}$-HO; [*Herskowitz and Jensen, 1991*]). Both yWS304 and yWS1626 were recovered from the *MAT***a** haploid genomic deletion library.

yWS1689 (*MAT***a** *his3Δ1 leu2Δ0 met15Δ0 ura3Δ0 ydj1::KAN^R rce1::KAN^R*) and yWS1693 (*MAT***a** *his3Δ1 leu2Δ0 met15Δ0 ura3Δ0 ydj1::KAN^R ste24::KAN^R*) were created by a similar approach. In these instances, yWS304 containing pSM703 was crossed to yWS1686 (*MATα his3 leu2 met15 ura3 rce1:: KAN^R*) and yWS1685 (*MATα his3 leu2 met15 ura3 ste24::KAN*), respectively. pSM703 contains a *URA3* marker used for diploid selection that was subsequently lost from the desired haploid by

**Table 1.** Strains used in this study.

| Strain Identifier | Genotype | Reference |
|---|---|---|
| IH1783; ATCC#204278 | *MAT*a *trp1 leu2 ura3 his4 can1* | (*Michaelis and Herskowitz, 1988*) |
| IH1793; ATCC#204279 | *MAT*α *lys1* | (*Michaelis and Herskowitz, 1988*) |
| SM1188; ATCC# 204273 | *MAT*a *trp1 leu2 ura3 his4 can1 ste14-3::TRP1* | (*Hrycyna et al., 1991*) |
| SM2331 | *MAT*a *trp1 leu2 ura3 his4 can1 mfa1-Δ1 mfa2-Δ1* | (*Chen et al., 1997*) |
| SM3689 | *MAT*a *trp1 leu2 ura3 his4 can1 mfa1-Δ1 mfa2-Δ1 rce1::TRP1* | (*Tam et al., 1998*) |
| yWS42; BY4741 | *MAT*a *his3Δ1 leu2Δ0 met15Δ0 ura3Δ0* | (*Shoemaker et al., 1996*) |
| yWS164 | *MAT*a *trp1 leu2 ura3 his4 can1 mfa1-Δ1 mfa2-Δ1 rce1::TRP1 ste24::KAN$^R$* | (*Cadiñanos et al., 2003*) |
| yWS304 | *MAT*a *his3Δ1 leu2Δ0 met15Δ0 ura3Δ0 ydj1::KAN$^R$* | (*Shoemaker et al., 1996*) |
| yWS1577 | *MAT*α *his3Δ1 leu2Δ0 met15Δ0 ura3Δ0 ydj1::KAN$^R$* | This study |
| yWS1626 | *MAT*a *his3Δ1 leu2Δ0 met15Δ0 ura3Δ0 ste14::KAN$^R$* | (*Shoemaker et al., 1996*) |
| yWS1629 | *MAT*a *his3Δ1 leu2Δ0 met15Δ0 ura3Δ0 rce1:: KAN$^R$* | (*Shoemaker et al., 1996*) |
| yWS1632 | *MAT*a *his3Δ1 leu2Δ0 met15Δ0 ura3Δ0 ram1::KAN$^R$* | (*Shoemaker et al., 1996*) |
| yWS1635 | *MAT*a *his3Δ1 leu2Δ0 met15Δ0 ura3Δ0 ste14::KAN$^R$ ydj1::KAN$^R$* | This study |
| yWS1682 | *MAT*a *his3Δ1 leu2Δ0 met15Δ0 ura3Δ0 ste24:: KAN$^R$* | (*Shoemaker et al., 1996*) |
| yWS1685 | *MAT*α *his3Δ1 leu2Δ0 met15Δ0 ura3Δ0 ste24:: KAN$^R$* | This study |
| yWS1686 | *MAT*α *his3Δ1 leu2Δ0 met15Δ0 ura3Δ0 rce1:: KAN$^R$* | This study |
| yWS1689 | *MAT*a *his3Δ1 leu2Δ0 met15Δ0 ura3Δ0 rce1::KAN$^R$ ydj1::KAN$^R$* | This study |
| yWS1693 | *MAT*a *his3Δ1 leu2Δ0 met15Δ0 ura3Δ0 ste24::KAN$^R$ ydj1::KAN$^R$* | This study |
| yWS2078 | *MAT*a *his3Δ1 leu2Δ0 met15Δ0 ura3Δ0 ydj1::KAN$^R$ [RNQ$^+$]* | (*Summers et al., 2009*) |
| yWS2109 | *MAT*a *his3Δ1 leu2Δ0 met15Δ0 ura3Δ0 YDJ1* | This study |
| yWS2110 | *MAT*a *his3Δ1 leu2Δ0 met15Δ0 ura3Δ0 YDJ1 (SASQ)* | This study |
| yWS2111 | *MAT*a *his3Δ1 leu2Δ0 met15Δ0 ura3Δ0 YDJ1 (CTLM)* | This study |
| yWS2112 | *MAT*a *his3Δ1 leu2Δ0 met15Δ0 ura3Δ0 YDJ1 (CVIA)* | This study |
| yWS2117 | *MAT*a *his3Δ1 leu2Δ0 met15Δ0 ura3Δ0 ste14:: KAN$^R$ YDJ1* | This study |
| yWS2118 | *MAT*a *his3Δ1 leu2Δ0 met15Δ0 ura3Δ0 ste14:: KAN$^R$ YDJ1 (SASQ)* | This study |
| yWS2119 | *MAT*a *his3Δ1 leu2Δ0 met15Δ0 ura3Δ0 ste14:: KAN$^R$YDJ1 (CTLM)* | This study |
| yWS2120 | *MAT*a *his3Δ1 leu2Δ0 met15Δ0 ura3Δ0 ste14:: KAN$^R$YDJ1 (CVIA)* | This study |

selection on *5-fluoroorotic acid*. yWS1685 and yWS1686 are *MAT*α derivatives of yWS1682 and yWS1626, respectively, which were retrieved from the *MAT*a haploid genomic deletion library.

yWS2109-yWS2112 were created by restoring the disrupted *ydj1Δ::Kan$^R$* locus of BY4741 (*ydj1Δ:: Kan$^R$*) to encode wildtype Ydj1p or *CaaX* variants SASQ, CTLM, and CVIA, respectively. In a similar manner, yWS2117-2120 were created by restoring the *ydj1Δ::Kan$^R$* locus of yWS1635 (*ste14Δ::Kan$^R$ ydj1Δ::Kan$^R$*). In each case, the integrating fragment was derived from a *CEN* plasmid bearing the appropriate *YDJ1* variant that was digested extensively with multiple restriction enzymes to destroy other plasmid encoded genes (*e.g. URA3*) but not *YDJ1* or its 5′and 3′ untranslated regions. Candidates with appropriate phenotypes (*i.e.* kanamycin-sensitive; uracil auxotrophy; growth at 37°C) were evaluated by diagnostic PCR for the presence of *YDJ1* and the absence of *ydj1Δ::Kan$^R$*. The sequence of the *YDJ1* open reading frame was confirmed by sequencing of PCR fragments derived from the *YDJ1* locus. Two or more isolates were typically validated.

## Plasmids

The plasmids used in this study are listed in *Table 2*. Plasmids were either previously reported or constructed by standard molecular methods. All plasmids used were analyzed by restriction digest and sequencing to verify the proper sequence of the entire open reading frame.

**Table 2.** Plasmids used in this study.

| Plasmid Identifier | Genotype | Reference |
| --- | --- | --- |
| pGAL-HO | CEN URA3 $P_{GAL}$-HO | (*Herskowitz and Jensen, 1991*) |
| pRS315 | CEN LEU2 | (*Sikorski and Hieter, 1989*) |
| pRS316 | CEN URA3 | (*Sikorski and Hieter, 1989*) |
| pRS415 | CEN LEU2 | (*Sikorski and Hieter, 1989*) |
| pRS416 | CEN URA3 | (*Sikorski and Hieter, 1989*) |
| pSM703 | 2μ URA3 $P_{PGK}$ | (*Zhang et al., 2001*) |
| pSM1093 | CEN URA3 STE24 | (*Fujimura-Kamada et al., 1997*) |
| pSM1194 | 2μ URA3 STE24 | (*Fujimura-Kamada et al., 1997*) |
| pSM1275 | CEN URA3 RCE1 | (*Schmidt et al., 1998*) |
| pSM1316 | CEN LEU2 STE14 | (*Romano and Michaelis, 2001*) |
| pWS270 | CEN URA3 $P_{GAL}$-GFP-RAS2 | (*Manandhar et al., 2007*) |
| pWS523 | CEN URA3 $P_{GAL}$-GST-YDJ1 | (*Zhu et al., 2001*) |
| pWS546 | CEN URA3 $P_{GAL}$-GFP-RAS2-CASQ | This study |
| pWS610 | CEN LEU2 MFA1 | (*Krishnankutty et al., 2009*) |
| pWS612 | CEN LEU2 MFA1-CASQ | (*Krishnankutty et al., 2009*) |
| pWS613 | CEN LEU2 MFA1-CTLM | This study |
| pWS882 | CEN URA3 $P_{GAL}$-GFP-YDJ1 | This study |
| pWS942 | CEN URA3 YDJ1 | This study |
| pWS948 | 2μ URA3 $P_{PGK}$-YDJ1 | This study |
| pWS972 | 2μ URA3 $P_{PGK}$-YDJ1-SASQ | This study |
| pWS1132 | CEN URA3 YDJ1-SASQ | This study |
| pWS1246 | CEN URA3 YDJ1-CTLM | This study |
| pWS1247 | 2μ URA3 $P_{PGK}$-YDJ1-CTLM | This study |
| pWS1286 | CEN URA3 YDJ1-CVIA | This study |
| pWS1291 | 2μ URA3 $P_{PGK}$-YDJ1-CVIA | This study |
| pWS1298 | 2μ URA3 $P_{PGK}$-GST-YDJ1 | This study |
| pWS1307 | 2μ URA3 $P_{PGK}$-His$_6$-YDJ1 | This study |
| pWS1326 | CEN LEU2 YDJ1 | This study |
| pWS1327 | CEN LEU2 YDJ1-SASQ | This study |
| pWS1328 | CEN LEU2 YDJ1-CTLM | This study |
| pWS1329 | CEN LEU2 YDJ1-CVIA | This study |
| pWS1389 | CEN URA3 GFP-YDJ1 | This study |
| pWS1390 | CEN URA3 GFP-YDJ1-SASQ | This study |
| pWS1391 | CEN URA3 GFP-YDJ1-CTLM | This study |
| pWS1392 | CEN URA3 GFP-YDJ1-CVIA | This study |
| pWS1430 | CEN URA3 $P_{Gal1}$-PrD | (*Summers et al., 2009*) |
| pWS1431 | CEN URA3 $P_{CUP1}$-PrD-GFP | (*Summers et al., 2009*) |

The low-copy **a**-factor expression plasmid pWS613 (*CEN LEU2 MFA1-CTLM)* was constructed essentially as previously described for pWS610 and pWS612 (*Krishnankutty et al., 2009*).

The low-copy Ydj1p expression plasmids were constructed as follows. pWS942 (*CEN URA3 YDJ1*) was derived by subcloning a *YDJ1*-encoding PCR fragment into the XhoI and BamHI sites of pRS416. The PCR fragment was amplified from yeast genomic DNA and contained both 5´ and 3´ untranslated regions associated with *YDJ1*; the fragment was designed to retain a naturally occurring

XhoI site 5′ of the *YDJ1* ORF and was engineered through oligo design to contain a BamHI site at the 3′ end of the fragment. pWS1132 (*CEN URA3 YDJ1-SASQ*) was derived by QuikChange such that a silent NheI site was introduced along with the *CaaX* motif mutation. pWS1246 (*CEN URA3 YDJ1-CTLM*) and pWS1286 (*CEN URA3 YDJ1-CVIA*) were derived by modifying pWS1132 through PCR-directed, plasmid-based recombination (*Oldenburg et al., 1997*). In brief, pWS1132 digested with *Nhe*I was co-transformed into yeast along with a PCR product having the desired sequence changes and sequence homology to the parent plasmid in regions flanking the restriction site(s) to allow for gap repair. Following co-transformation, plasmids recovered from yeast colonies surviving SC-Ura selection were evaluated by restriction enzyme mapping and sequencing to confirm the identity of the desired plasmid. *CEN LEU2* versions of the above plasmids were constructed by subcloning *Xho*I-*Bam*HI fragments into the same sites of pRS315.

The multi-copy Ydj1p expression plasmids were constructed as follows. pWS948 (*2μ URA3 P_{PGK}-YDJ1*) was derived by subcloning a *YDJ1*-encoding PCR fragment into the EcoRI and NotI sites of pSM703. The PCR fragment was amplified from yeast genomic DNA and lacked the untranslated regions associated with *YDJ1*; the fragment was engineered to contain EcoRI and NotI sites adjacent to the 5′ and 3′ ends of the *YDJ1* ORF. pWS972 (*2μ URA3 P_{PGK}-YDJ1-SASQ*) was derived by PCR-directed, plasmid-based recombination involving pWS948 as the recipient plasmid and a PCR fragment generated using a megaprimer site-mutagenesis approach. The PCR fragment was amplified from pWS948. pWS1247 (*2μ URA3 P_{PGK}-YDJ1-CTLM*) and pWS1291 (*2μ URA3 P_{PGK}-YDJ1-CVIA*) were constructed by subcloning the *YDJ1*-encoding BamHI-BsaBI fragments from pWS1246 and pWS1286, respectively, into the same sites of pWS948.

The His_6-Ydj1p expression plasmid (pWS1307) was constructed in two steps. First, a PCR fragment encoding *GST-YDJ1* was introduced into pSM703 by PCR-directed, plasmid-based recombination to yield pWS1298 (*2μ URA3 P_{PGK}-GST-YDJ1*). The fragment was amplified from pWS523 (*CEN URA3 P_{GAL}-GST-YDJ1*) using appropriate mutagenic oligos; pWS523 was recovered from an arrayed collection of yeast containing over-expression plasmids for individual yeast ORFs (*Zhu et al., 2001*). Next, a fragment encoding a poly-histidine tag was introduced into pWS1298 such that the GST sequence and a linker region were eliminated.

The GFP-Ydj1p expression plasmids (pWS1389-1392) were each constructed by subcloning a PCR fragment, designed to encode GFP flanked by *YDJ1* 5′ non-coding and coding sequences, into appropriate Ydj1p expression plasmids (pWS942, pWS1132, pWS1246 and pWS1286, respectively). This resulted in GFP encoded at the 5′ end of the *YDJ1* open reading frame. The DNA fragment was produced by PCR from pWS882 (*CEN URA3 P_{GAL}-GFP-YDJ1*) with appropriately designed primers, digested with *Eco*RI and *Pfl*MI, and ligated into the same sites on recipient plasmids. pWS882 itself was created by PCR-directed, plasmid-based recombination involving the introduction of a PCR fragment encoding *YDJ1* into pWS270 (*CEN URA3 P_{GAL}-GFP-RAS2*) (*Manandhar et al., 2007*). The recombination event replaced *RAS2* with *YDJ1*.

The GFP-Ras2p CASQ expression plasmid (pWS546) was created using mutagenic oligos and PCR-directed, plasmid-based recombination using pWS270 (*CEN URA3 P_{GAL}-GFP-RAS2*) as the parent plasmid (*Manandhar et al., 2007*). The recombination event altered the *CaaX* encoding sequence from CIIS to CASQ.

## Temperature sensitivity assay

Strains were cultured to saturation (25°C, 24–30 hr) in appropriate liquid media. Plasmid-bearing strains were cultured in SC-Ura, and yeast with integrated copies of *YDJ1* variants were cultured in YPD. Cultures were then serially diluted into YPD using a multi-channel pipettor and a 96-well plate (10-fold dilutions); the first well of each series contained the undiluted culture. Each serially diluted culture was spotted onto YPD solid media (5 μl per spot), and the plates incubated at 25°C, 37°C, or 40°C. To ensure reproducibility, incubator temperatures were continuously monitored using digital thermometer probes. The images of plates were digitized with a flat bed scanner after an appropriate time of growth to yield similar sized colonies for the Ydj1 (CASQ) strain: 25°C for 72 hr; 37°C for 48 hr; 40°C for 72 hr plus 48–72 hr at non-restrictive temperature to allow better visualization of micro-colonies. Each experiment was performed at least twice on separate days, and each strain was evaluated in duplicate within each experiment.

## Over-expression growth assays

Strains for the plate-based assay were cultured to saturation in SC-Ura,Leu liquid media, normalized, and serially diluted as described for the temperature sensitivity assay, except that selective media was used for dilution series and spotting of strains. Plates were incubated at 30°C and scanned after 3 days. Each experiment was performed at least twice on separate days, and each strain was evaluated in duplicate within each experiment.

Strains for the doubling-time assay were cultured to saturation in SC-Ura,Leu liquid media, diluted with fresh media ($A_{600} \sim 0.15$), and growth monitored over 24 hr at 30°C in a 96-well plate format using a Synergy HT microplate reader. $A_{600}$ readings were taken every four minutes with shaking between readings (i.e. mixing for 150 s). The $A_{600}$ measurements were analyzed using Prism (Graph-Pad Software, La Jolla, CA) to generate doubling times using time segments of exponential growth (see *Figure 3—source data 1*). Each strain was evaluated in quadruplicate.

## Yeast lysate preparations for SDS-PAGE

Yeast strains expressing Ydj1p were cultured to log phase ($A_{600}$ 0.5–1.0) in selective SC-Ura at 25°C unless otherwise noted. Cell pellets of equal mass were harvested by centrifugation, washed with water, and processed by alkaline hydrolysis and TCA precipitation (*Kim et al., 2005*). Samples were directly resuspended in urea-containing Sample Buffer prior to analysis by SDS-PAGE and immuno-blot. For GFP-Ras2p studies, strains were first cultured in SRaffinose/Glycerol-Ura liquid media to mid-log then supplemented with 2% Galactose for 5 hr prior to harvesting the cells.

## SDS-PAGE, Immunoblot, and image analysis

Samples along with PageRuler size standards (ThermoScientific) were separated by SDS-PAGE (12.5 or 15%), transferred onto nitrocellulose, and blots processed for immunoblotting according to standard protocols and published methods (*Caplan and Douglas, 1991*). In brief, the blot was blocked with non-fat dry milk (5% w/v) resuspended in TBST (100 mM Tris, 400 mM NaCl, 0.1% Tween 20, pH 7.5), incubated with appropriate dilutions of rabbit anti-Ydj1p primary antibody (courtesy of Dr. Avrom Caplan) and HRP-conjugated donkey anti-rabbit secondary antibody (GE Healthcare Cat# NA934 RRID:AB_772206), and immune complexes detected on X-ray film after treatment of blot with HyGLO development solution (Denville Scientific, South Plainfield, NJ). The developed film image was digitized (300 dpi) using a flat-bed scanner. Where GFP-Ras2p and PrD-GFP were analyzed, mouse anti-GFP (Abnova Corporation Cat# MAB9749 RRID:AB_10750945) was the primary antibody and HRP-conjugated sheep anti-mouse was the secondary antibody (GE Healthcare Cat# NA931 RRID:AB_772210).

## Ydj1p purification

Yeast strains were cultured in selective SC-Ura liquid media at 25°C to log phase ($A_{600}$ 0.5–1.0). Cell pellets were harvested by centrifugation, washed with water, then incubated on ice in 100 mM Tris (pH 9.4) containing 10 mM DTT for 10 min at a density of 20 $A_{600}$/ml. Cell pellets were recovered, resuspended in Lysis Buffer B (50 mM HEPES, pH 7.4, 300 mM NaCl, 15% glycerol, 5 M urea), and subject to mechanical lysis by bead beating (4 cycles of 4 min; 4°C). Lysates were clarified by two rounds of centrifugation (1000 xg, 10 min, 4°C) before batch purification of poly-histidine tagged proteins using Talon resin (Clontech, Palo Alto, CA). After incubating the lysate with resin for 30 min at 25°C, the resin was washed twice with cold lysis buffer B containing 5 mM imidazole, and bound protein eluted using Lysis Buffer B containing 1 M imidazole.

## Mass spectrometry analysis

The purified $His_6$-Ydj1p sample was denatured by incubating with 10 mM dithiothreitol at room temperature for an hour, alkylated with 55 mM iodoacetamide for 45 min in the dark, and digested with endoproteinase GluC (Promega, Madison, WI) overnight at 37°C in 40 mM ammonium bicarbonate. The resulting peptides were cleaned up using C18 spin columns (The Nest Group, Southborough, MA), dried down, and reconstituted in 0.1% formic acid. The peptides were separated on a 75 μm (I. D.) x 15 cm C18 capillary column (packed in house, YMC GEL ODS-AQ120ÅS-5, Waters, Milford, MA) and eluted into the nano-electrospray ion source of an Orbitrap Fusion Tribrid mass spectrometer (Thermo Fisher Scientific, Waltham, MA) with a 180-min linear gradient consisting of 0.5–100%

solvent B over 150 min at a flow rate of 200 nL/min. The spray voltage was set to 2.2 kV and the temperature of the heated capillary was set to 280°C. Full MS scans were acquired from m/z 300 to 2000 at 120 k resolution, and MS2 scans following collision-induced fragmentation were collected in the ion trap for the most intense ions in the Top-Speed mode within a 3-s cycle using Fusion instrument software (v1.0, Thermo Fisher Scientific, Waltham, MA). The raw spectra were searched against the *Saccharomyces cerevisiae* database (UniProt, RRID:SCR_002380; March 2015, 7376 entries) using SEQUEST (Proteome Discoverer 1.4, Thermo Fisher Scientific, Waltham, MA) with full MS peptide tolerance of 20 ppm and MS2 peptide fragment tolerance of 0.5 Da, and filtered at the peptide level to generate a 1% false discovery rate for peptide assignments and manually validated for the farnesylated peptide.

## GFP-Ydj1p and GFP-Ras2p microscopy

Yeast cultured to mid-log ($OD_{600}$ between 0.6–0.8) in liquid SC-Ura media were harvested by centrifugation and concentrated 5-fold in fresh media. For GFP-Ras2p studies, cells were induced in SGal-Ura for 6 hr prior to use as previously described (*Manandhar et al., 2007*). The cell suspensions were spotted onto a microscope slide, and fluorescent images captured using a Zeiss (Oberkochen, Germany) Axioplan microscope equipped with Plan-Neofluar 100×/1.30 NA oil immersion objective lens, fluorescence optics, and Zeiss Axiocam MRc charge-coupled device camera. Image capture was controlled using AxioVision software in RGB mode with approximately 1 s exposure times. Digital files were imported into Photoshop CS3, and select images were imported into PowerPoint for figure construction. Fields of cells were scored for the association of cells with puncta (see *Figure 4— source data 1*). Only cells with moderate to strong GFP signals were scored to minimize false negatives.

## Yeast mating assays

Serial-dilution yeast mating assays were performed as described previously (*Kim et al., 2005*; *Alper et al., 2006*). In brief, strains were cultured to saturation in appropriate liquid media. *MAT*a strains were cultured in SC-Leu or SC-Ura,Leu as appropriate. The *MAT*α strain (IH1793) was cultured in YPD. Saturated cultures were diluted with fresh media to an $A_{600}$ value 0.95 ± 0.05. The normalized *MAT*a cell suspensions were individually mixed 1:9 with the *MAT*α cell suspension in wells of a 96-well plate (160 μl final volume). The mating mixtures were then serially diluted into three additional wells into volumes of the *MAT*α cell suspension using a multi-channel pipette (10-fold dilutions; 160 μl final volume). Each diluted series was spotted (5 μl per spot) in duplicate onto solid minimal (SD) and selective media lacking lysine (SC-K). Colonies were counted on plates after 4 days of incubation at 30°C. Growth of diploid colonies on SD is indicative of mating events. Growth patterns on SC-K were used to confirm accurate dilution of the MAT**a** cells. Digitized images of plates were recorded using a flatbed scanner (300 dpi), and imported into Photoshop for image adjustment (*i.e.* rotation and contrast optimization).

Quantitative mating assays were performed by spreading diluted mating mixtures (50 μl) on SD and SC-K plates in duplicate. In each case, the mixtures were further diluted with *MAT*α cell suspension to insure a suitable volume for spreading (*i.e.* 150–200 μl). The dilution chosen for spreading onto SD was empirically determined but was generally the second highest dilution to yield mating events in the serial-dilution mating test. The dilution chosen for spreading onto SC-K was always the most diluted sample in the series (*i.e.* 4th well). Colonies were counted after 4 days of incubation at 30°C, and the ratio of diploid colonies recovered on SD plates (*i.e.* mating events) relative to the number of colonies recovered on the SC-K plates (*i.e.* input number of *MAT*a cells) was determined. These values were used to calculate relative mating efficiencies that were either expressed as a percentage or fold difference.

## PrD proteotoxicity and co-immunoprecipitation assays

[*RNQ*+] *ydj1Δ yeast* (yWS2078) were constructed to contain an inducible PrD over-expression plasmid (pWS1430) or constitutive PrD-GFP expression plasmid (pWS1431) in combination with a low-copy Ydj1p-expression plasmid (pWS1326-1329) or empty vector (pRS315). PrD over-expression strains were cultured in liquid selective media to saturation, serially diluted, and spotted as for temperature sensitivity assays, except that dilutions were into $H_2O$, spots (7.5 μl) were onto SC-Ura/Leu

and SGal-Ura/Leu, and incubations were at 30°C. PrD-GFP strains were cultured to mid-log, and lysates prepared as previously described (*Summers et al., 2009*). In brief, cells were mechanically broken by vortexing with silica beads in Buffer A (50 mm HEPES, pH 7.4, 150 mm NaCl, 0.1% Triton X-100) containing 1 mM phenylmethylsulfonyl fluoride and 1x protease inhibitor mixture (Research Products International, Mount Prospect, IL), crude lysates clarified by centrifugation (1000 xg, 5 min), and clarified samples assayed for total protein concentration. A portion of each lysate (10 μg total protein) was analyzed directly by SDS-PAGE and immunoblot with either Ydj1p or GFP antibody. An additional portion (100 μg total protein) was incubated with Ydj1p antibody and immune complexes recovered using Protein A Sepharose (GE Healthcare Life Science, Pittsburg, PA). Immune complexes were analyzed by SDS-PAGE followed by immunoblot with GFP antibody. Immunoprecipitated Ydj1p could not be visualized because it was obscured by the presence of the antibody heavy chain that co-migrated with Ydj1p on the gel.

## Sequence alignments

Sequences and alignment scores were retrieved from the HomoloGene (RRID:SCR_002924) database using "Ydj1" as the query (http://www.ncbi.nlm.nih.gov/homologene). The retrieved lists were culled to remove over-representation of mammalian sequences and multiple entries from the same organism. For the latter, the sequence with the highest identity score relative to Ydj1p was retained. The last 25 amino acid positions of the culled sequences were analyzed by WebLogo (RRID:SCR_010236) to determine the amino acid frequency at each position (*Crooks et al., 2004*).

## Acknowledgements

We thank A Caplan (City College of New York) for Ydj1p antiserum, D Cyr (University of North Carolina School of Medicine) for the PrD-encoding plasmid and [*RNQ+*] strain, M Momany (University of Georgia; UGA) for fluorescence microscope access, SP Manandhar (Schmidt lab, UGA) for assistance with GFP-Ydj1p studies, MC Samuelson-Ruiz (Schmidt lab, UGA) for assistance with yeast mating studies, and members of the Schmidt and Wells labs (UGA) for critical discussions. This work was financially supported in part by NIGMS/NIH P41GM103490 (LW) and funds provided by the University of Georgia (WKS).

## Additional information

### Funding

| Funder | Grant reference number | Author |
| --- | --- | --- |
| National Institute of General Medical Sciences | P41GM103490 | Lance Wells |
| University of Georgia | | Walter K Schmidt |

The funders had no role in study design, data collection and interpretation, or the decision to submit the work for publication.

### Author contributions

ERH, Conception and design, Acquisition of data, Analysis and interpretation of data, Drafting or revising the article, Contributed unpublished essential data or reagents; MC, JHK, Acquisition of data, Analysis and interpretation of data; PZ, Acquisition of data, Analysis and interpretation of data, Drafting or revising the article; LW, Analysis and interpretation of data, Drafting or revising the article; WKS, Conception and design, Acquisition of data, Analysis and interpretation of data, Drafting or revising the article

### Author ORCIDs

Lance Wells, http://orcid.org/0000-0003-4956-5363
Walter K Schmidt, http://orcid.org/0000-0002-3359-3434

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
