## [Decision Letter]

Thank you for submitting your article "A shunt pathway limits the *CaaX* processing of Hsp40 Ydj1p and regulates Ydj1p-dependent phenotypes" for consideration by *eLife*. Your article has been favorably evaluated by Vivek Malhotra as the Senior editor and three reviewers, one of whom is a member of our Board of Reviewing Editors.

The reviewers have discussed the reviews with one another and the Reviewing Editor has drafted this decision to help you prepare a revised submission.

Summary:

This study investigates the C-terminal processing of a heat shock protein Ydj1p by farnesylation and the protease and carboxymethylation steps thereafter. Primarily relying on genetic methods in yeast, the authors show that the natural C-terminal sequence CASQ of Ydj1p in yeast is farnesylated but not proteolyzed or carboxymethylated. They show that Ydj1p farnesylation is critical for function (based on results with the SASQ mutant) but that forced proteolysis and carboxymethylation by substituting with C-terminal sequences from a-factor or Ste18p disrupt Ydj1p function as evidenced by thermotolerance differences. They also show that CASQ processing is context dependent (i.e. different with protease overexpression and presence in a-factor). The work appears carefully done and rigorous and is the first clear demonstration that prenylation without further processing is a biological requirement for at least one protein.

Essential revisions:

What is not clear from the manuscript is why further processing is counterproductive for Ydj1p and whether there are for example, any 'readers' for a specific C-terminal code. We think the manuscript would be significantly strengthened if the authors attempted to investigate whether CASQ vs. CVIA versions of Ydj1p showed differences in binding partners. This could involve testing by immunoprecipitation whether there are changes in the degree of association with GDI, Yip or other possible interacting proteins.

Another limitation of the study is the possibility that overexpression of Ydj1p constructs might alter the efficiency of processing, perhaps by swamping out the key membrane proteases, and thereby influence the phenotypes and levels of processing observed. We think this could be addressed in at least two ways. One of the ways to address this would be to use chromosomally inserted forms of the *CXXX* Ydj1p and then assess the thermotolerance of the strains. An alternative approach would be to perform mass spectrometry on an overexpressed form of Ydj1p with a CVIA C-terminal sequence (same plasmid and promoter as the Figure 5 experiment; rce1 KO). In this case, one would be testing the prediction that overexpressed Ydj1p-CVIA was completely processed by Ste24p in contrast to the result with Ydj1p-CASQ which was not under these conditions.

---

## [Author Response]

*Essential revisions:*

*What is not clear from the manuscript is why further processing is counterproductive for Ydj1p and whether there are for example, any 'readers' for a specific C-terminal code. We think the manuscript would be significantly strengthened if the authors attempted to investigate whether CASQ vs. CVIA versions of Ydj1p showed differences in binding partners. This could involve testing by immunoprecipitation whether there are changes in the degree of association with GDI, Yip or other possible interacting proteins.*

We agree that our data does not clarify this important issue. The possibility that the natural Ydj1p *CaaX* motif interacts with specific ‘readers’ is an intriguing possibility. We first investigated, as suggested, whether interactions between Ydj1p *CaaX* mutants and specific client proteins were disrupted. But, it was unclear to us why investigations of GDI and Yip1 were promoted, since we are not aware of evidence (genetic, physical, published) that these proteins interact with Ydj1p. Instead, we chose to investigate the interaction between Ydj1p and the prion domain of Rnq1p (PrD), which is known to be dependent on farnesylated Ydj1p. We confirmed the previously reported farnesyl-dependency but did not observe altered interactions for Ydj1p *CaaX* mutants despite using multiple technical approaches, including co-IP and genetic analyses. These results are now reported as an additional figure (Figure 7). We also investigated the interaction between Ydj1p and the glutamine-rich fragment of Htt and observed no differences with *CaaX* mutants. Lastly, we evaluated expression of Axl1p whose mRNA and protein levels are dependent on Ydj1p (Meacham et al., J. Biol. Chem. 1999, 274:34396-34402). We observed reduced levels of Axl1p in the absence of Ydj1p (*i.e. ydj1*∆) but not for any of the Ydj1p *CaaX* mutants, including Ydj1p SASQ. The Htt and Axl1p results are mentioned in the text but not reported in figure format. Of the considerable number of other genetic and physical interactors reported for Ydj1p (over 400 unique interactors on the *Saccharomyces* Genome Database), we expect that a case-by-case approach could eventually yield an appropriate “reader”. But, extensive case-by-case analyses and/or more appropriate global analyses would have quickly expanded our effort beyond the scope and timeframe of this current study. In our opinion, our negative results for interaction studies do not weaken our conclusion that the shunt pathway is functionally relevant for certain *CaaX* proteins.

*Another limitation of the study is the possibility that overexpression of Ydj1p constructs might alter the efficiency of processing, perhaps by swamping out the key membrane proteases, and thereby influence the phenotypes and levels of processing observed. We think this could be addressed in at least two ways. One of the ways to address this would be to use chromosomally inserted forms of the CXXX Ydj1p and then assess the thermotolerance of the strains. An alternative approach would be to perform mass spectrometry on an overexpressed form of Ydj1p with a CVIA C-terminal sequence (same plasmid and promoter as the Figure 5 experiment; rce1 KO). In this case, one would be testing the prediction that overexpressed Ydj1p-CVIA was completely processed by Ste24p in contrast to the result with Ydj1p-CASQ which was not under these conditions.*

We now report that integrated forms of Ydj1p produce thermotolerance phenotypes similar to those observed in plasmid-based expression studies. Moreover, the rescuing effect of a *ste14*∆ deletion was again observed as reported with plasmid-based studies. We interpret these findings to indicate that observed phenotypes reflect underlying alterations in physiology that are not due to the mild over-expression of Ydj1p achieved with a *CEN*-based plasmid system.